# Monitoring Immune Responses in Neuroblastoma Patients during Therapy

**DOI:** 10.3390/cancers12020519

**Published:** 2020-02-24

**Authors:** Celina L. Szanto, Annelisa M. Cornel, Saskia V. Vijver, Stefan Nierkens

**Affiliations:** 1Center for Translational Immunology, University Medical Center Utrecht, Utrecht University, 3584 CX Utrecht, The Netherlands; c.l.szanto-2@prinsesmaximacentrum.nl (C.L.S.); A.M.Cornel@umcutrecht.nl (A.M.C.); s.v.vijver@students.uu.nl (S.V.V.); 2Princess Máxima Center for Pediatric Oncology, Utrecht University, 3584 CS Utrecht, The Netherlands

**Keywords:** Neuroblastoma, biomarker, immunotherapy, anti-GD2, Dinutuximab, immune profiling, immune monitoring, adoptive cell therapy, checkpoint inhibitors, cytokines

## Abstract

Neuroblastoma (NBL) is the most common extracranial solid tumor in childhood. Despite intense treatment, children with this high-risk disease have a poor prognosis. Immunotherapy showed a significant improvement in event-free survival in high-risk NBL patients receiving chimeric anti-GD2 in combination with cytokines and isotretinoin after myeloablative consolidation therapy. However, response to immunotherapy varies widely, and often therapy is stopped due to severe toxicities. Objective markers that help to predict which patients will respond or develop toxicity to a certain treatment are lacking. Immunotherapy guided via immune monitoring protocols will help to identify responders as early as possible, to decipher the immune response at play, and to adjust or develop new treatment strategies. In this review, we summarize recent studies investigating frequency and phenotype of immune cells in NBL patients prior and during current treatment protocols and highlight how these findings are related to clinical outcome. In addition, we discuss potential targets to improve immunogenicity and strategies that may help to improve therapy efficacy. We conclude that immune monitoring during therapy of NBL patients is essential to identify predictive biomarkers to guide patients towards effective treatment, with limited toxicities and optimal quality of life.

## 1. Introduction

Neuroblastoma (NBL) is a tumor derived from sympathoadrenal progenitor cells of the developing sympathetic nervous system. It occurs most often in the adrenal medulla or sympathetic ganglia [1,2,3]. NBL is the most commonly diagnosed solid tumor during the first year of life and is responsible for approximately 15% of pediatric cancer deaths [2,3,4]. Risk classification of patients is based on different clinical factors, such as patients’ age and International Neuroblastoma Risk Group (INRG) tumor stage, as well as biologic factors, such as histopathologic classification, DNA ploidy, MYCN status, and copy-number of chromosome 11q [3]. Outcome dramatically differs between patients with different tumor stages.

The high-risk NBL tumor environment is often referred to as ‘cold’ or ‘immune-deserted’, characterized by presence of very few immune cells in the tumor microenvironment (TME) [5]. However, the probability for a cold tumor to respond to immune therapy depends on strategies to transform it to ‘hot’ tumors [6]. The cold phenotype is likely caused by the development of multiple immunomodulatory mechanisms by the tumor and its environment, including major histocompatibility complex I (MHC-I) downregulation, regulatory T cell (Treg) and myeloid-derived suppressor cell (MDSC) accumulation, and decreased T cell cytotoxicity [7]. Infiltrating immune cells are observed especially in low-risk NBL. The presence of tumor-infiltrating lymphocytes (TILs) was found to be correlated with favorable clinical outcome [8]. This suggests a role of immune infiltration in regression of NBL, which is supported by increased serum levels of granulysin, an effector molecule of cytotoxic T cells, observed in a case study of spontaneous NBL regression [9]. Therapeutic interference to increase immune infiltration and recognition might therefore be key to increase therapy efficiency against NBL. 

High-risk NBL is currently treated with surgery, radiotherapy, 5–8 cycles of intensive chemotherapy, including platinum-, alkylating-, and topoisomerase agents—which is often followed by autologous stem cell transplantation (ASCT)—and immunotherapy [1,2,3,4,10]. High expression of GD2 (a disialoganglioside) across NBLs and low expression levels in healthy tissue has led to the rationale of GD2 targeting immunotherapy [11]. Administration of the chimeric monoclonal antibody (mAb) anti-GD2 (ch14.18), combined with the cytokines IL-2 and granulocyte macrophage-colony stimulation factor (GM-CSF), and isotretinoin in patients with high-risk NBL resulted in a significant increase 2-year event-free (EFS) and overall survival (OS) [10]. The observation of this effect, despite the harsh immunomodulatory immune environment of NBL, shows the potential of immune interference in NBL. However, as about 40% relapse is still observed in these patients, there is a clear medical need to optimize (immuno)therapeutic strategies. The current immunotherapy protocol is particularly ineffective for high-burden disease. In addition, osteomedullary metastatic disease occurs in most patients with high-risk neuroblastoma [12]. Elucidating the mechanisms of effective anti-tumor responses is key to find out, and act upon, what discriminates responders from non-responders.

Several studies in multiple types of cancer report increased tumor infiltration of immune cells upon (chemo)therapy [13,14,15], which could potentially predict overall therapy response and prognosis in an early stage. Immune monitoring during therapy provides the opportunity to study biological mechanisms of response and resistance [16]. This enables identification of biomarkers to monitor therapy response, potentially aiding to early stratification of responders and non-responders. In the 1960s, it was reported that the correlation between prognosis and degree of lymphocyte infiltration is also observed in NBL [17,18,19]. It is now known that NBL tumors are intermixed with different immune cells, recently identified to include CD4- and CD8-T cells, natural killer (NK) cells, and γδ-T cells [20]. Interestingly, Mina et al. showed that the prognostic value of TIL levels at diagnosis is even better than criteria currently used to stage NBL, such as MYCN amplification [8]. This illustrates the potential role of immune cells in influencing the clinical outcome and emphasizes the need for standardized immune monitoring during therapy in this patient group.

This review provides an overview of predictive immune biomarkers of clinical response to treatment, emphasizes the importance of immune monitoring during NBL treatment, and describes its relevance for evaluation of the immune response and patient stratification by developing new biomarkers. 

## 2. Immune Monitoring at Diagnosis: Correlates of Outcome?

### 2.1. Immune Markers at the Tumor Site 

Histological analysis of human tumor types, including melanoma, ovarian-, head- and neck, breast-, urothelial-, colorectal-, lung- hepatocellular-, and esophageal cancer showed the presence of tumor infiltrating immune cells, such as macrophages, dendritic cells (DCs), polymorphonuclear cells, NK cells, B cells, and T cells. Although these studies revealed a broad interpatient variability, a high density of CD3 T cells, CD8 T cells, and CD45RO+ memory T cells was generally associated with improved EFS and OS [21,22]. In addition, for NBL, multiple studies show that increased CD3 T cell infiltration and proliferation is associated with favorable clinical outcome [8,23]. These data correspond with the findings that high-risk MYCN-amplified primary metastatic NBL tumors show lower levels of infiltrated lymphocytes, monocytes, and macrophages, and exhibit lower interferon pathway activity and chemokine expression [23]. T cell proliferation is most likely impaired by high arginase activity in the TME [24], resulting in low arginine levels (an essential molecule for T cell proliferation). 

CD4 T cell infiltration is associated with better survival, regardless of MYCN amplification [23]. However, extensive phenotyping of these CD4 T cells, with markers such as CD25, CD127, and FoxP3 to distinguish regulatory T cells (Treg), is lacking. Based on gene set enrichment analyses, gene expression of IL-4 (indicative for Th2) was elevated and associated with better prognosis in tumors with high CD4 T cells infiltration whereas no association was observed with interferon γ (IFNγ), IL-2 and tumor necrose factor α (TNFα) (indicative for Th1) [25]. In addition, NKT cell infiltration has been reported to be favorable for outcome, possibly by inhibition of suppressive monocytes in the TME [26]. 

Presence of immune cell populations with presumed regulatory properties, including tumor associated monocytes, macrophages (TAMs), and Tregs, predict poor outcome [8,27]. Tumor infiltrating macrophages often display an immunosuppressive M2-phenotype supporting T cell suppression, tumor cell migration, and treatment evasion [28]. High expression of TAM-associated genes CD14, CD16, IL-6, IL-6R, and transforming growth factor (TGFβ1) is associated with decreased 5-year EFS [26]. Additionally, monocytes isolated from the TME are able to suppress T cell proliferation in vitro [24] and excrete multiple soluble T cell inhibitory factors, such as TGFβ and IL-10 [29,30]. Although Tregs and MDSCs are well known for their suppressive effects on the immune system, associations of these subsets with clinical outcome remain limited to studies investigating bone marrow (BM) or peripheral blood (PB). One study showed that tumor-induced overexpression of high-mobility group box 1 (HMBG1) induces a Treg phenotype. Patients with overexpression of this protein were at higher risk for progression of disease, relapse, and death [7]. 

Despite these correlations in retrospective analyses, there are no prognostic markers in the TME to steer clinical decision making. Inclusion of markers of cell differentiation, function, activation, and exhaustion in multiple parameter analysis may help determine which factors have the strongest associations with, and prognostic value for patient outcome. Advanced techniques, such as tissue cytometry by time-of-flight (CyTOF), can overcome the limitations of measuring only a few markers in common practice immune histology. 

### 2.2. Circulatory Immune Markers 

Despite advances in immune profiling and the easy accessibility of, PB; no validated circulatory immune biomarkers exist for patients with NBL. Surprisingly, studies implementing immune monitoring in NBL patients are limited. 

#### 2.2.1. Cytokines and Soluble Molecules in Plasma/Serum 

Cytokines and chemokines are components of a complex network promoting angiogenesis and metastasis, diminishing adaptive immunity, and changing responses to hormones and therapeutic agents. As such, cytokines involved in cancer-related inflammation are easy to monitor and relate to patient outcome and could be a target for therapeutic strategies.

Oliveira and colleagues reported an increase of IL-2, IL-4, IL-5, IL-6, IL-9, IL-10, IL-13, IL-17A, IL-17F, IL-21, IL-22, interferon γ (IFNγ), and tumor necrose factor α (TNFα) in plasma of NBL patients compared to age matched controls. One of these cytokines, IL-6, is a key growth-promoting and anti-apoptotic inflammatory cytokine, which correlated with poor prognosis and high-risk disease [31]. When integrating levels of IL-6 with other candidate biomarkers (serum amyloid A (SAA), apolipoprotein (APOA1), epidermal growth factor (EGF), macrophage derived chemokine (MDC), sCD40L, and Eotaxin), the multivariate classifier predicted active disease with a sensitivity of 81% and specificity of 90% [32]. These data are encouraging and are awaiting validation in other patient cohorts.

Even though mRNA levels of IL-10 (a product of Tregs, NK cells and macrophages) in BM and, PB; as well as IL-10 plasma concentrations were higher in metastatic NBL patients compared to healthy controls, a prognostic role of IL-10 alone could not be demonstrated [29]. Low levels of soluble proteins other than cytokines, such as human leukocyte antigens HLA-E and HLA-F, were associated with worse prognosis [33]. In contrast to this, several other studies were not able to correlate markers measured in plasma/serum to outcome.

In other cancers, the analyses of cytokine profiles rather than single markers has shown positive prediction values at diagnosis [34,35,36,37]. The studies summarized above hopefully initiate validation studies in NBL patients with multiparameter analyses in different patient cohorts. 

#### 2.2.2. Immune Cells in Peripheral Blood 

Leukocyte counts have found to be significantly higher in NBL patients compared to controls [38]. Although no difference is observed in total lymphocyte count between healthy controls, localized, and metastatic patients, relative numbers of multiple lymphocyte subsets do vary [31]. Semeraro et al. measured a significant increase in the percentage of CD3−CD56+ NK cells in PB of metastatic NBL patients compared to patients with localized tumors, which was associated with a minor response to induction chemotherapy. The percentage of cytotoxic (CD16+) NK cells positively correlated with clinical response to therapy [39]. These correlations could potentially be explained by NK cell mediated cytotoxic effects on MHC-I lacking tumor cells. Activated NK cells may also upregulate MHC-I expression on NBL cells, thereby circumventing further NK cell mediated cytotoxicity, while at the same time increasing their susceptibility to T cell mediated cytotoxicity [40]. 

Morandi and colleagues found that Treg (CD4+CD25hiCD127−) and Tr1 (CD4+CD45R0+CD49b+LAG3+) subsets are decreased in NBL patients compared to controls, but no correlation was found with prognostic factors, such as age and stage. MYCN amplification was the only prognostic factor associated with higher levels of Treg numbers in BM and Tr1 levels in PB [41]. In addition, CD4+ and CD8+ T cells show increased surface expression levels of the checkpoint inhibitor CTLA-4, and PD-1 on CD4 T cells. In contrast, Semeraro et al. found increased CD4+FoxP3+ T cells in metastatic NBL patients compared to localized tumors. The differences in the markers used to identify specific cell subsets (in this case Treg) in different studies complicates a valid comparison between the data from these studies and indicates the need for harmonization of immune phenotyping protocols. 

When comparing the myeloid compartment, NBL patients with localized tumors showed higher monocyte, neutrophil and erythrocyte counts as compared to patients with metastatic disease. When zooming in on the phenotype of myeloid cells, increased levels of the checkpoint inhibitor programmed death ligand 2 (PD-L2) were observed in transitional (CD14+CD16+) and non-classical monocytes (CD14-CD16+) in patients compared to controls [31]. Furthermore, increased expression of CSF-1R, a regulator inducing MDSC expression, was observed in patients, and correlated with poor clinical outcome [42]. 

In summary, it is clear that NBL patients show alterations in absolute numbers and subset percentages of immune cells, as well as in immune proteins in the TME and in PB. An overview of the reviewed studies can be found in Table 1. However, so far, no robust prognostic marker correlating with survival has been identified and validated. Such immune signatures at diagnosis could aid in therapy decision making and prognosis prediction. 

## 3. Immune Monitoring during Therapy 

Immune monitoring during therapy is crucial to identify potentially prognostic factors that could be exploited to enhance immunogenicity of the tumor and predict treatment response. This section will start by reviewing studies which monitor the immune response upon standard therapy, including chemotherapy and monoclonal antibody therapy. Subsequently, immune monitoring in more experimental treatment regimens, including vaccination strategies, adoptive cell-, and checkpoint inhibition therapy will be discussed.

### 3.1. Chemotherapy 

In general, high-risk NBL patients receive 5–8 cycles of intensive chemotherapy including platinum, alkylating, and topoisomerase agents. In North-America, induction regimens include vincristine, doxorubicin, cyclophosphamide, cisplatin, and etoposide, while the Society of pediatric oncology Europe NBL group (SIOPEN) used a rapid COJEC regimen that gives eight cycles with combinations of vincristine, carboplatin, etoposide, cyclophosphamide, and cisplatin [47].

A limited number of studies has monitored immune profiles during chemotherapy in cancer patients. Monitoring lymphocyte levels during and after chemotherapy in hematopoietic and solid tumors generally showed increased EFS in patients with higher lymphocyte counts at diagnosis as well as after induction chemotherapy [42,47,48,49] In addition, fast monocyte recovery after chemotherapy is predictive for EFS in patients with leukemias and lymphomas [50,51]. In line with these data, an elevated neutrophil to lymphocyte ratio after chemotherapy, but before surgical resection of the NBL tumor, was associated with decreased OS [52]. 

Upon chemotherapy treatment, Treg counts decreased, possibly due to nonspecific targeting of Tregs by chemotherapeutic agents. More studies are warranted to determine if the effect can be subscribed to chemotherapy-induced decrease in T cells in general, or whether specific subsets, like Tregs, might be more susceptible to chemotherapy-induced cytotoxicity [53]. Chemotherapy generally does not affect NK cells [54], however, the expression levels of NKp30, an NK cell receptor involved in tumor cell killing and DC recognition, positively correlate with survival after chemotherapy [55]. Expression levels of the immunosuppressive isoform NKp30C and the activating isoforms NKp30A and NKp30B affect NK cell function and correlate with EFS of NBL patients after chemotherapy [39]. 

No studies have monitored immune markers in the TME during chemotherapy in patients. An in vivo mouse study showed that depletion of TAMs from NBL tumors is associated with increased chemotherapeutic efficacy without requiring T cell contribution [56]. This observation led to the author’s suggestion to combine CSF-1R blockade with chemotherapy to potentially increase treatment efficacy. 

To date, too few studies have monitored immune status during chemotherapy to identify markers that could predict response to therapy. It seems that patients with higher numbers or faster recovery of lymphocytes and monocytes have better EFS changes. Whether this is a reflection or a result of the development of a healthier immunological niche should be studied, but the observation could lead to the hypothesis that these patients might also be responding better to immune treatment options.

### 3.2. Monoclonal Antibody Therapy 

A frequent immunotherapy protocol of high-risk NBL consists of anti-GD2 combined with all-trans-retinoic acid (ATRA), IL-2, and GM-CSF. Even though immunotherapy increased 2-year EFS and OS [10], relapse is still observed in 40% of patients. Elucidation of effective anti-tumor responses is key to study what discriminates responders from non-responders. 

IL-2 and GM-CSF have been added to the treatment protocol as they were observed to enhance cytotoxicity of anti-GD2 in vitro [10,57,58,59]. In addition, GM-CSF also increased myeloid cell activation, another important cell type in the anti-tumor response [57,60]. The multi-component nature of the immunotherapy protocol makes that the observed immune effects cannot be ascribed to a specific component of the protocol.

A primary mechanism of action of anti-GD2 is the induction of antibody-dependent cellular cytotoxicity (ADCC), which requires recognition by effector cells (mainly NK cells, monocytes, neutrophils, and macrophages) [11,60]. Cytotoxic activity of NK-cells is mediated by CD16, whereas cytotoxic activity of monocytes, neutrophils, and macrophages is mediated by CD32. Both receptors recognize the Fc fragment of anti-GD2 on opsonized NBL cells and induce cytotoxic effector functions. Complement-dependent cytotoxicity (CDC) is another mechanism of action of anti-GD2, however, most studies focus on its implications regarding pain toxicity rather than on-tumor toxicity. 

Nassin et al. monitored immune reconstitution at the start of immunotherapy containing IL-2, GM-CSF, and anti-GD2 [61]. They showed that absolute lymphocyte counts (T., B.; and NK subsets) are lower in the vast majority of patients as compared to age-matched controls. Patients with disease progression, relapse or residual disease had significantly lower total leukocyte counts, as well as a lower absolute lymphocyte-, neutrophil-, and CD16+ cell counts compared to disease-and progression-free patients observed three months after therapy. Siebert and colleagues found that presence of human anti-chimeric antibodies against chimeric anti-GD2 resulted in significant reduction in peripheral anti-GD2 levels, as well as significant abrogation in ADCC and CDC [62]. However, in this study, it is not clear if such immune responses are a disadvantage for survival of the treated patients. In addition, the induction of the host anti-idiotype network, measured indirectly by human anti-mouse antibody responses correlated with long term survival [63,64,65] 

The importance of NK cells in ADCC was illustrated by Chowdhury et al.; showing that in vitro anti-GD2 mediated lysis of the LAN1 NBL cell line upon co-culture with peripheral blood mononuclear cells (PBMCs) from a NBL patient abrogated after NK cell depletion [66]. In addition, cell lysis correlated with NK cell expression of CD69, an early activation marker, as well as with the degranulation marker CD107a. Furthermore, variation in ADCC between patients was found to be caused by genetic predispositions resulting in better cytotoxic activity of effector cells and correlations with better survival [59,63,64]. Siebert et al. studied the level of ADCC in vitro in combination with FCGR polymorphisms and killer cell immunoglobulin like receptor KIR/KIR ligand genotypes of 53 patients. They showed that patients with high affinity FCGRs had higher ADCC levels and better EFS compared to patients with low affinity genotypes. In addition, a correlation was found between the activating KIR 2DS2 genotype on ADCC and EFS. A combination of high-affinity FCGR2A,-3A and stimulating genotypeB/x or the presence of activating KIR 2DS2 resulted in the strongest anti-NBL cellular cytotoxicity mediated by anti-GD2 and improved EFS [67]. In addition, Tarek et al. found that patients treated with monoclonal antibodies (moABs) lacking HLA class I ligands for their inhibitory KIRs have significantly higher survival rates. Unlicensed NK cells mediate tumor control via ADCC [68]. These results show that FCGR polymorphisms and KIR/KIRL genotypes could function as biomarkers in response to immunotherapy.

The importance of NK cell mediated ADCC in anti-GD2 efficacy, together with the observation of relatively fast NK cell recovery early after ASCT was an important rationale for immunotherapy timing early after transplantation [69]. However, more detailed evaluation of NK cell subsets showed that most of these NK cells are immature, cytokine releasing (CD56bright, CD16+/−) rather than the cytotoxic (CD56dim, CD16+) NK cells known to be mainly responsible for anti-GD2 dependent ADCC [61]. This is further substantiated by observed impaired immune recovery of CD16+ NK cells in patients with disease progression or relapse at time of transplantation compared to those without. These studies may suggest suboptimal timing of anti-GD2 immunotherapy early after transplantation. Utilizing haploidentical allogeneic hematopoietic cell sources rather than autologous sources could be explored as a transplantation source regarding NK cell recovery early after transplantation.

In addition to NK cell subset monitoring, Nassin et al. showed increased CD25 expression on CD4 T cells as compared to CD8 T cells at the start of immunotherapy [61]. Even though FoxP3 expression of these cells is unknown, it is hypothesized that these cells could be classified as Tregs [70]. Ladenstein et al. recently concluded from a phase III clinical trial that there is no additive effect of IL-2 administration on outcome of high-risk NBL patients [71]. As (low dose) IL-2 administration to autoimmune patients resulted in preferential expression of Tregs [72], authors hypothesize that adjuvant IL-2 administration could be responsible for Treg expansion during immunotherapy, diminishing the positive effects (e.g., NK expansion) of IL-2. Indeed, we observed a rise in Treg numbers upon every round of IL-2 to NBL patients (unpublished). Furthermore, administration of IL-2 also results in development of eosinophils through stimulating growth factors derived from T cells, such as IL-3, IL-5, and GM-CSF that help to maintain eosinophils in vitro [73]. In addition, an increase in IL-5 levels during immunotherapy with anti-GD2 and cytokines was observed, that could induce and maintain eosinophils [74]. Although their effect is unknown, suppressive eosinophils have been reported in murine studies and could possibly also be present in IL-2 treated NBL patients [75]. Further studies are required to confirm these effects of IL-2 on levels of Treg and eosinophils, preferably with functional testing of the different cell subsets.

To overcome limitations of the current anti-GD2 monoclonal antibody therapy, including monoclonality of the response, anti-idiotype responses, and memory induction, Kusher et al. reported a strategy in which patients are vaccinated with GD2 and GD3. This results in an intrinsic, polyclonal, multivalent antibody response through stimulation of B-cells to produce anti-GD2 and –GD3 [76]. As B-cell recovery is key for vaccination efficacy, immunomonitoring of B-cell recovery after ASCT is key for optimal vaccination timing.

To conclude, immune monitoring studies during immunotherapy are largely lacking and studies that are available have only monitored patients at start and end of therapy. All new phase II/III trials and standard treatment protocols should not only asses outcome but also monitor the immune system during therapy to get a better and faster understanding of treatment success.

### 3.3. Adoptive Cell Therapy 

The development of adoptive cell therapy (ACT) strategies has taken flight in the last decade. These cell products should ideally possess the capacity to expand, actively migrate through the entire body (including to the solid tumor core and over the blood brain barrier), and induce systemic immune memory to prevent future relapse. ACT products are generated by harvesting, ex vivo expansion, and re-direction of immune cells to target tumor cells. Even though successes have been achieved in several hematological malignancies [77,78,79,80], translation of these successes to solid tumors is difficult. Target expression heterogeneity, localization to the tumor site, and overcoming the immunosuppressive, nutrient-, and stimuli- deprived TME are thought to be the main challenges in effective adoptive cell therapy in solid tumors [81]. 

To date, pre-clinical studies as well as clinical trials are exploiting T-, NK-, and NKT cells in both autologous and allogeneic ACT strategies in NBL. Infused cells can be isolated from PB [82,83,84,85], as well as from tumor tissue (e.g., TILs) [8,20]. In addition, isolated cells can be genetically modified to improve recognition of tumor cells, for example through knock-in of a NBL-specific T cell receptor (TCR) [40,86,87], a chimeric antigen receptor (CAR) [20,88,89,90,91,92,93,94,95], or a bispecific antibody [96,97] against tumor specific targets such as GD2, PRAME, NY-ESO-1, L1-CAM, B7-H3, and mutated ALK. Very low or even absent surface expression of major histocompatibility complex I (MHC-I) on NBL cells has led to the focus on MHC-I unrestricted ACT-strategies, mainly exploiting NK cell- and CAR therapy (or a combination of both). Excellent overviews of clinical trials of (adoptive cell) therapy strategies in NBL are provided by Le and Thai [98] and Zage [99]. 

Even though multiple recent (pre-clinical) studies have demonstrated efficacy of various forms of ACT in NBL, the observed clinical benefit is limited. Persistence of ACT products in general is a widely discussed topic and is thought to be depending on balanced activation (appropriate co-stimulation, prevention of tonic/chronic receptor signaling, and of activation-induced cell death), tolerance induction (caused by native antigen expression and tumor immunosuppression), the cell phenotype of both the apheresis material and the product itself, as well as by the need of antigen availability for cell persistence [87,100,101]. The extremely immunosuppressive, nutrient- and stimuli deprived TME of solid tumors is thought to be the main factor responsible for the discrepancies in successes with hematological tumors [82,102]. 

Monitoring the tumor infiltrating cell population, its phenotype, and other related factors will provide the much-needed insight to predict therapy response and to understand what is driving resistance and success and where can be acted upon. Sporadic monitoring after ACT resulted in the observation that the presence of central memory and naïve T cell phenotypes in cell products has a positive effect on persistence of the cell product in several cancer types, including NBL [8,82,103,104,105,106]. Hurtado et al. recently showed that the presence of central memory NBL TILs greatly decreases after non-specific ex vivo rapid expansion cycles, stressing the importance of ex vivo expansion protocol evaluation in adoptive cell strategies [20]. Moreover, lymphodepletion before T cell infusion or early T cell infusion after ASCT (day 2) caused significantly improved expansion and persistence of the cell product [82,89]. This indicates that the immunosuppressive status of the immune system of these patients is an important limiting factor in effective ACT and led to the rationale to combine ACT with checkpoint inhibitors [89,107]. Remarkably, combining PD-1 blockade by pembrolizumab with GD2 CAR T cell administration upon lymphodepletion in a phase 1 clinical trial in NBL did not show any beneficial effect of PD-1 blockade on peripheral CAR T cell expansion [89]. A limitation of this study is that the arm studying the effect of PD-1 blockade on non-lymphodepleted GD2 CAR T cell treated patients was missing. Furthermore, T cell expansion and persistence was solely measured in the blood and not at the tumor site. The fact that two out of three treated patients in this arm experienced complete remission is encouraging and warrants more research. Controversially, immune monitoring in the same phase 1 GD2 CAR study showed specific expansion of circulating immunosuppressive M2 macrophage-like myeloid cells (CD45+CD33+CD11b+CD163+) independent of lymphodepletion and PD-1 blockade [89]. Inhibitory myeloid cells are correlated with poor prognosis in several cancer types, including NBL, even though this study does not provide any data on whether these mobilized cells are attracted to the tumor site. Induction of MDSCs was also reported in GD2−CAR T cell therapy for sarcomas in xenograft mice, which impaired CAR T cell activity [108]. Addition of all-trans retinoic acid (ATRA) destroyed the MDSCs and thereby improved the efficacy of the GD2-CAR T cells [108], suggesting that patients might also benefit from ATRA to eradicate the circulating M2 macrophage-like myeloid cells that are induced by GD2-CAR3. Thirdly, immune monitoring in this phase 1 study indicated a correlation between circulating IL-15 levels and GD2 CAR T cell expansion [89]. Circulating IL-15 levels could not only potentially be used as a biomarker for CAR T cell expansion, but a pre-clinical xenograft mouse model transduced with GD2 CAR T cells overexpressing IL-15 also showed improved expansion, enhanced anti-tumor activity and improved survival [93].

Even though immune monitoring data during ACT in NBL is scarce, the above mentioned studies all resulted in clear rationales for research into new therapeutic strategies or biomarker development. This clearly indicates the need for more elaborate immune monitoring during ACT. 

### 3.4. Checkpoint Inhibitors

Another interesting strategy would be to target the immunosuppressive environment of NBL [7]. Checkpoint inhibitors (CPis), blocking CTLA-4, PD-1, PD-L1, or PD-L2, are proven to be effective in a variety of tumors, including solid tumors [109]. The potential of these inhibitors is illustrated by the growing list of FDA-approvals, as reviewed by Hargadon et al. [110]. Currently, most of these approvals are in unresectable metastatic disease settings. Nevertheless, the potency of CPis to induce a more potent anti-tumor immune response, and therefore potential induction of anti-tumor immunological memory, makes checkpoint inhibition a very interesting candidate as an adjuvant therapy in curative treatment settings. The utilization of CPis as a monotherapy in NBL has been investigated in multiple pre-clinical studies [42,111,112,113]. These studies show no effect of CPi treatment on systemic NBL progression in vivo. This has been supported in a phase I clinical trial in pediatric patients with advanced solid tumors, in which one NBL patient was included [114]. More clinical trials assessing CPis as a monotherapy in refractory NBL are currently running (NCT02304458; NCT02332668; NCT02541604). 

Several rationales have been proposed for the ineffectiveness of CPi therapy in high-risk NBL. First of all, evaluation of NBL tumors from patients with different tumor stages does not only, as mentioned before, show inverse correlation between tumor grade and TILs [8], but also between tumor grade and PD-L1 tumor expression [113]. Absence of TILs and PD-L1 tumor expression provides a first rationale to CPi therapy resistance. Secondly, one of the immunomodulatory mechanisms of NBL is the upregulation of PD-L1 in response to IFN-γ [111,113]. This highlights a potential resistance mechanism against CPi therapy, as persistent IFN-γ production by activated T cells may lead to a continuous cascade causing even higher upregulation of the immune checkpoints resulting in T cell senescence. Impaired IFN pathway activity in MYCN-amplified tumors [23] may prevent PD-L1 expression upon IFN-γ exposure and provides a third rationale for CPi therapy effectiveness in high-risk NBL. 

Next to these observations, especially studies investigating CPi-including combination therapies revealed some insights in mechanisms explaining NBL resistance to CPi monotherapy [42,111,112,113]. Rigo and colleagues showed that combining CPi with temporary CD4 depletion by anti-CD4 monoclonal antibody (mAb) treatment caused a very potent CD8 T cell dependent response causing significantly longer tumor-free survival, complete tumor regression, and durable anti-NBL immunity in vivo [111]. Combining CPi with immune enhancer IL-21, or Treg targeting agents polyoxotungstate (POM-1) and anti-CD25 antibody, revealed no significant effect. This indicates that CD4 CD25high Tregs, as well as adenosine generation in Tregs are not responsible for CPi resistance in NBL. More specific studies into effects of CD4 CD25−/low FOXP3+ Tregs or other CD4 Tregs should be performed to further unravel the synergistic effect observed upon CD4 depletion [111]. 

Mao et al. reported that targeting of CSF-1Rþ+ suppressive myeloid cells in combination with CPis caused spontaneous control of NBL in vivo [42], a mechanism confirmed in several other types of cancers, as reviewed by Weber et al. [115]. A possible explanation for this is the observation that T cells start to produce M-CSF upon PD-1 blockade, which can bind to CSF-1R on myeloid-derived suppressor cells, thereby enhancing their suppressive phenotype [116]. This causes a further reduction of IFN-regulated chemokine release (e.g., CXCL9, 10, and 11) in the TME, which are important for T cell infiltration and could therefore potentially explain the synergistic effect of targeting CSF-1Rþ+ suppressive myeloid cells in combination with CPi. 

Other combination strategies which seem promising based on preclinical studies include combinations of CPis with whole tumor cell vaccination [113], high intensity ultrasound [117], and anti-GD2 treatment [112]. All mentioned preclinical evidence of effective NBL targeting treatment combinations provides a clear rationale for clinical assessment of CPi therapy as an adjuvant therapy against NBL. Even though data obtained in these induced tumor mouse models should be interpreted with caution, the data should be used as a rationale for focusing on immune monitoring of adjuvant CPi therapy in coming clinical trials. 

## 4. Discussion 

The present review shows the potential of standardized immune monitoring in NBL. Despite many efforts, high-risk NBL has a poor response to treatment and novel therapies are urgently required. We should therefore better understand the functional kinetics of tumor- and immune cells to guide and modulate immune therapy strategies. As such, immune monitoring may provide evidence-based directions to optimize treatment protocols, e.g., in the recent discussion whether IL-2 should be removed from the current treatment protocol [118].

Studies listed in this review use flow cytometry, CyTOF, immunohistochemistry, as well as functional assays to perform immune monitoring on tissue, PBMCs or plasma/serum from NBL patients. Monitoring PBMCs and plasma/serum cytokines is by far the easiest approach as blood is easily available during treatment. As the immune system is a multifaceted system, it is important to question if findings in the circulatory system are related to the TME. Monitoring peripheral blood subsets currently have insufficient clinical value for clinical decision making. Future studies with paired monitoring of PB, BM, and TME samples will increase our understanding of underlying mechanism and shed light on the use of biomarkers over the treatment course as predictors for efficacy and/or toxicity.

As most studies describe small patient populations, high interpatient variability, and no harmonization of monitoring protocols, it can be difficult to interpret results and relate it to recent literature [119]. The studies used in this review are sometimes limited to results of 3–5 patients. Therefore, we propose to define and implement standardized immune monitoring protocols in the trial- and study design to increase patient numbers and study interpatient variability. An example is proposed in Figure 1. As each patient is unique, immune monitoring can facilitate future personalized treatment in NBL. 

Advances in organoid technology is promising in predicting the immunosuppressive capacities as well as the sensitivity to (immune) therapy in a personalized setting. Neuroblastoma tumor models have been developed that represent the tumor better than classical cell lines [120]. This strategy has previously shown to be effective in metastatic gastrointestinal cancers, in which patient derived organoids were instrumental in the prediction of tumor specific responses, stimulating personalized treatment strategies [121]. Single-cell RNA sequencing of NBL tumors can, besides studying tumor heterogeneity and driving factors behind tumor heterogeneity, help to identify new leads for immunotherapeutic strategies [122].

## 5. Conclusions

In conclusion, immune monitoring before and during therapy of NBL patients could facilitate identification of predictive biomarkers to guide patients towards effective treatment with limited toxicities and optimal quality of life. Furthermore, understanding the immunomodulatory environment of NBL and its response to treatment in responders and non-responders is important to facilitate design of new therapeutic strategies improving outcome of high-risk NBL. 

## Figures and Tables

**Figure 1 cancers-12-00519-f001:**
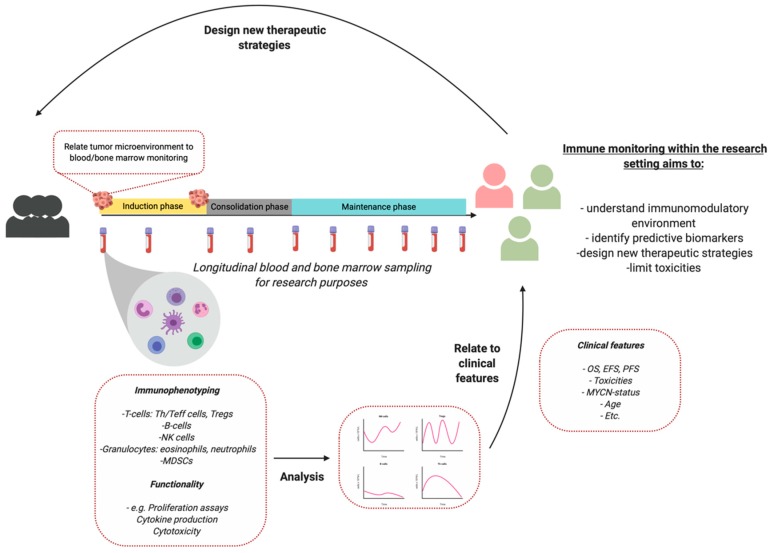
Immune monitoring for research purposes. Monitoring peripheral blood subsets currently have insufficient clinical value for clinical decision making. *Created with BioRender.com.*

**Table 1 cancers-12-00519-t001:** Overview of neuroblastoma immune monitoring studies at diagnosis.

Flow Cytometry				
Number of Unique Patient Samples Measured Including Material		Tumor Characteristics	Markers	Reference
8	Primary Tumor	I (4x) + II (1x) + III (2x) + IV (1x)	CD3, CD4, CD8, CD25, CD45RA, CCR7	Carlson et al. 2013 [43]
	PB			
26	Primary Tumor	II (2x) + III (2x) + IV (21x) + IVs (1x) 15 MYCN amp, 11 non MYCN amp	GD2	Mussai et al. 2015 [24]
	PB		CD15, CD14, CD11b	
20	BM	IV (20x)	CD45, CD33, CD14, GD2, CD56	Song et al. 2009 [26]
41	PB	13 MYCN amp, 28 non MYCN amp	CD4, CD25, CD127, CD45RO, CD49b, LAG-3	Morandi et al. 2015 [29]
5	PB	High-Risk Patients	HLA-DR, CD33, CD11b	Gowda et al. 2013 [44]
21	PB	7 MYCN amp, 20 non MYCN amp (14 localized, 13 metastatic)	CD4, CD25, CD127, CD45RO, CD49b, LAG-3	Morandi et al. 2016 [41]
27	BM			
59	PB	23 localized, 36 metastatic tumors	CD8, NKp46, CD4, CD16, CD56, NKp30, DNAM-1, CD127, CD25, CD14, CD45, CD15, GD2, CD235a, CD9, CD81, TCRgd, NKp44, NKp80, CD3e, CD158a/h, CD158b, CD158e/k, CD158i, FoxP3	Semeraro et al. 2015 [39]
	BM			
**Immunohistochemistry**				
24	Primary Tumor	7 MYCN amp, 26 non MYCN amp	CD68	Apps et al. 2013 [45]
21			CD3	
19			pSTAT3	
8	Primary Tumor	I (4x) + II (1x) + III (2x) + IV(1x)	Ki67, CD3	Carlson et al. 2013 [43]
15	Primary Tumor	3 low risk, 6 intermediate risk, 6 high risk	CD4, CD45	Zhang et al. 2017 [25]
129	Primary Tumor	IV (129x)	CD1d, Vα24-Jα18inv, TCRα6β11	Song et al. 2017 [26]
71	Primary Tumor	stage I–III (n = 29), stage IV (n = 31), stage IVS (n = 11)	CD163, AIF1	Asgharzadeh et al. 2012 [27]
84	Primary Tumor	I (34x) + II (19x) + III (5x) + IV (20x) + IVS (6x)	CD3, CD4, CD8, CD25, FOXP3, Ki67, β2m-free MHC1 heavy chain	Mina et al. 2015 [8]
**Elisa**				
57	Plasma	IV (49x) + non IV (8x)	IL-10, ARG-1 (57)	Morandi et al. 2015 [29]
53	PB	PB: I (8x) + II (8x) + III (6x) + IV (28x) + IVS (3x)	IL-6	Egler et al. 2008 [46]
18	BM	BM: I (1x) + II (3x) + III (2x) + IV (11x) + IVS (1x)		
35	PB	PB: I (5x) + II (5x) + III (3x) + IV (20x) + IVS (2x)	sIL-6R	
16	BM	BM: I (1x) + II (2x) + III (2x) + IV (10x) + IVS (1x)		
84	Primary Tumor	I (7x) + II (8x) + III (22x) + IV (42x) + 4S (5x) 27 MYCN amp, 57 non MYCN amp	sHLA-F, sB7H3, sHLA-E	Morandi et al. 2013 [33]
**Luminex**				
55	Plasma	20 low-risk, 35 high-risk In addition, 28 HR blood samples from 7 patients at various timepoints during treatment	GM-CSF, G-CSF, IFNγ, IL-1a, IL-1ra, IL-1b, IL-2, IL-3, IL-4, IL-5, IL-6, IL-7, IL-8, IL-9, IL-10, IL-12p70, IL-12p40, IL-13, IL-15, IL-17, MCP-1, MCP-3, MDC, TNFα, TNFβ, TGFα, Eotaxin, IFNα2, IP-10, MIP-1a, MIP-1b, EGF, FGF-2, FLT3L, Fractalkine, GRO, VEGF, sCD40L, sIL-2Ra	Egler et al. 2011 [32]
	PB			

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
