# Peer review of "Monitoring Immune Responses in Neuroblastoma Patients during Therapy"

_cancers, 2020, doi:10.3390/cancers12020519_

Round 1
Reviewer 1 Report
This is a good attempt at summarizing and reviewing a subject which has not been well studied.
Major concerns:
Although the authors mention NB being a "cold" tumor, this should be put into context and expanded upon. NB, like most pediatric tumors is not amenable to T-cell mediated therapies due to several reasons including lack of HLA class I expression. The reader should not be given the impression that current T-cell therapies are at all effective and that a major advance is needed for such therapies to be beneficial for NB (and other pediatric tumors). Introduction: Ab-mediated anti-GD2 immunotherapy is considered standard of care. However, it is relatively ineffective for high burden disease. This should be made clear. The unmet need in improving outcomes for NB is osteomedullary metastatic disease. Thus, evaluating microenvironment of solid masses might have limited value. This unmet need should also be put into context. Peripheral blood subsets are NOT useful for immune monitoring: this should be emphasized. Immune monitoring as it relates to observations in patients: anti-idiotype network should be discussed: note refs: Cheung etal Cancer 1995, Kushner etal Oncoimmunology 2015. The connection to antiganglioside vaccine should be discussed. Kushner etal 2014. NK cells: the concept of licensed vs nonlicensed and role of KIR receptors should be explored: Tarek etal 2012 Allotransplant approached using haploidentical approach should be explored as related to NK cell function Figure 1. The presented data (and available data in the literature) do not support routine immune monitoring in the clinic. Figure 1 is likely to give the reader the impression that this should be considered using current technology. However, even based on the authors’ presentation, current methods do not yield useful information. Such an approach can be considered for research purposes to answer specific questions, but such an approach constitutes a fishing expedition and waste of resources given the current state of knowledge. Authors should consider discussing/speculating newer methods of immune monitoring e.g. receptor repertoire, newer methods of evaluating tumor microenvironment.
Several references are incomplete etc
Author Response
Please see the attachment, page 2,3 and 4.

Reviewer 2 Report
Szanto et al. present a thorough overview of immune monitoring in neuroblastoma patients and highlight the need for its use after the establishment of standard protocols with the aim of using this information for more effective treatments.
Despite the complexity of the topic, the review is well-written and well-structured. It provides the necessary background information and points out the next steps that need to be taken to advance the field.
Minor points that need to be adressed:
Some acronyms are not explained (i.e. SAA, APOA1, MDC, HCT…). Because of the presence of numerous acronyms, authors could provide an initial list of them.
In the paragraph 3.2 “Monoclonal Antibody Therapy”, Authors should explain more about the results found by Siebert et al., not just give a brief information.
Some typos need to be corrected:
lines 137-138: the meaning is not clear in “as well as an easy to monitor parameter for patient outcome”
and
line 276: references are in different brackets
Author Response
Please see the attachment, page 5 and 6.

Reviewer 3 Report
In this manuscript, Szanto C.L. and colleagues provided an overview regarding the role of immunotherapy in Neuroblastoma (NBL) focusing on the immune monitoring during NBL treatment and the development of new biomarkers. Overall, the paper is well written and clear in the study rational. However, in my opinion there are few minor issues/comments to be addressed before the manuscript can be accepted.
- It could be interest to investigate the application of organoid technology in immunotherapy in NBL
There are literature data/findings about that?
-Improve the layout and the organization of Table 1.
Author Response
Please see the attachment, page 7.

Round 2
Reviewer 1 Report
Responses to critiques 5, 10, 11 have not been incorporated into the new version.
